# 2D Leaf-Like Structured ZIF-L Embedded Electrochemically Reduced Graphene Oxide Composite as an Electrochemical Sensing Platform for Sensitively Detecting Benomyl

**DOI:** 10.3390/molecules27206857

**Published:** 2022-10-13

**Authors:** Min Shi, Guanwei Peng, Shuya Xue, Jingkun Xu, Yansha Gao, Shuwu Liu, Xuemin Duan, Limin Lu

**Affiliations:** 1Flexible Electronics Innovation Institute (FEII), Jiangxi Science and Technology Normal University, Nanchang 330013, China; 2Key Laboratory of Crop Physiology, Ecology and Genetic Breeding, Ministry of Education, Key Laboratory of Chemical Utilization of Plant Resources of Nanchang, College of Science, Jiangxi Agricultural University, Nanchang 330045, China

**Keywords:** two-dimensional MOFs, zeolitic imidazolate framework, ERGO, benomyl, electrocatalysts, electrochemical sensor

## Abstract

In this work, a two-dimensional leaf-like framework-L embedded electrochemically reduced graphene oxide (ERGO@ZIF-L) was proposed as an outstanding electrode material for the sensitive electrochemical sensing of benomyl (BM). ZIF-L is surrounded by ERGO, which could effectively ensure the stability and dispersion of ZIF-L. With this unique combination, the prepared ERGO@ZIF-L displayed excellent synergistic characteristics with a large surface area, excellent conductivity, plentiful active sites, and high electrocatalytic properties, thus endowing it with high sensitivity for BM determination. The experimental parameters, such as solution pH, material volume, and accumulation time, were optimized. Under optimal conditions, the BM sensor showed a wide linear range (0.009–10.0 μM) and low-limit detection (3.0 nM). Moreover, the sensor displayed excellent stability, repeatability, and reproducibility, and good anti-interference capability. The method was successfully applied to detect BM in real-world samples.

## 1. Introduction

Benomyl (BM) is one of the most widely used benzimidazole fungicides; it is used in the prevention and control of crop diseases and pests, and the preservation of vegetables and fruits [1,2]. Although it can effectively improve the quality of agricultural products, a large amount of BM remaining in the soil will seriously endanger human health and the ecological environment [3,4]. Thus, simple, reliable, and rapid methods for BM detection should be developed. A variety of methods have been established for the analysis of BM, including high-performance liquid chromatography, fluorescence polarization, fluorimetry, and electrochemical methods [5,6,7,8,9]. Electrochemical methods are particularly attractive because they possess multiple advantages; they are cost-effective and easy to operate, and have excellent sensitivity and good selectivity [10]. Within the electrochemical sensors, the electrode material plays a critical role in the efficient detection of BM [11].

Metal-organic frameworks (MOFs) are a class of porous crystalline materials consisting of metal ions with organic linkers [12], which have been applied in hydrogen storage, adsorption, catalysis, and sensors for their tunable morphology and large specific surface area [13,14,15,16]. Recently, MOF nanomaterial, especially the two-dimensional leaf-like framework-L (ZIF-L), has received considerable attention because it can be exploited in electrochemical applications [17,18]. The large surface area of ZIF-L can promote the concentration of analytes to high levels. The functional sites in ZIF-L, such as open metal sites, Lewis acidic/basic sites, and tunable pore sizes, can specifically recognize analytes using host–guest interactions or size constraints [19,20]. In addition, the leaf-like morphology can shorten ion diffusion and the electron transfer length, leading to an improvement in electrochemical performance [21,22]. Despite these recognized qualities, ZIF-L also has some inherent shortcomings, such as poor conductivity and easy agglomeration [23], which restricts its application.

It has been reported that hybrid MOFs with conductive graphene can effectively solve these issues [24,25]. At present, many methods have been proposed to prepare graphene oxides (RGO), such as chemical reduction, solvothermal reduction, photocatalytic reduction, and chemical vapor deposition [26,27,28,29]. Among these methods, electrochemical reduction has several advantages, such as a simple preparation process, a highly controllable degree of reduction, and no reaction by-products. For example, Wang et al. [30] prepared Cu-MOF/electrochemically reduced graphene oxide (ERGO) composite using one-step electrodeposition, where the Cu-MOF/GO composite was cast on glassy carbon electrode (GCE) and the GO was reduced by cyclic voltammetry. In our previous work [31], we reported a three-step method for the synthesis of ERGO-encapsulated Ce-BTC. Ce-BTC and GO composites were cast on the pre-cleaned electrode, followed by the electrochemical reduction of GO. The prepared ERGO-MOF composites presented a high specific surface area and good electronic conductivity, and demonstrated a satisfactory sensing. Compared with other types of MOFs, 2D ZIF-L has a unique structure and electronic characteristics [32]. This suggests that the combination of ERGO and ZIF-L could be a promising sensing electrode material.

Taking these issues into account, the ERGO@ZIF-L nanohybrid was first prepared through the electrostatic assembly of GO and ZIF-L, followed by the electrochemical reduction of GO to ERGO. The obtained ERGO@ZIF-L was ultimately proposed as an electrochemical sensor for BM detection. The high specific surface area of the ZIF-L nanosheets provided a large number of binding sites, which allowed for the high-efficiency enrichment of BM. Meanwhile, the excellent conductivity of ERGO significantly amplified the current signal. Accordingly, as an electrochemical sensor, ERGO@ZIF-L exhibited good selectivity, high reproducibility, low-limit detection, and a wide linear range. This prepared sensor was then successfully used to detect BM in tomatoes.

## 2. Results and Discussion

### 2.1. Structural Characterization

SEM was employed to investigate the morphologies of ZIF-L nanosheets, ERGO, and ERGO@ZIF-L. As seen in Figure 1A, numerous randomly-stacked leaf-like flakes with a relatively uniform size distributions (around 3–5 μm in length, 2 μm in width) were obtained, indicating that ZIF-L nanosheets were successfully prepared [17]. As shown in Figure 1B, the ERGO exhibited a typical wrinkled structure. For ERGO@ZIF-L (Figure 1C), a large quantity of the ZIF-L nanosheets was not only uniformly anchored on the ERGO surfaces, but was also intercalated into the layers of the ERGO sheets.

As can be seen in Figure 1D, the crystallographic structure of GO, ERGO, ZIF-L and ERGO@ZIF-L was confirmed via XRD analysis. The GO spectrum showed a major peak at 2θ values of 10.9°, which is the characteristic peak of GO [33]. However, after the electrochemical reduction, a feature diffraction peak at 2θ = 25.8° was found. The results suggested that the successful reduction of GO to ERGO can be obtained through the potentiostat technique [31]. For ZIF-L, diffraction peaks at 7.4°, 10.3°, 12.6°, 15.4°, 17.3°, 18.1° and 21.9° were assigned to the (200), (020), (220), (312), (023), (420) and (314) planes, respectively, which is consistent with previous research [19]. As for ERGO@ZIF-L, the XRD patterns revealed two characteristic peaks of ERGO and the main diffraction peak of ZIF-L, suggesting that ZIF-L nanosheets anchor onto the ERGO surface and do not alter the structure of the ZIF-L crystals. This further confirmed that ERGO@ZIF-L composites were successfully prepared.

The structure of the composite was further studied using Raman spectra. As can be seen in Figure 1E, two broad peaks were observed at 1345 and 1582 cm^−1^, corresponding to the D and G bands of carbon, respectively. Moreover, the position of the 2D-band (2722 cm^−1^) can be used to determine the number of graphene layers [34]. The intensity ratio between bands D and G (I_D_/I_G_) could exactly reflect the degree of graphitization in the carbon materials. The I_D_/I_G_ intensity ratio was 0.9987 for the ERGO, indicating that ERGO has a good graphite structure and fewer structural defects on its surface [35]. For ERGO@ZIF-L, the I_D_/I_G_ intensity ratio was significantly decreased to 0.9235, which suggested that ZIF-L nanosheets on the surface of the ERGO occupy the ERGO defect position. Furthermore, the 2D-band that appeared at ERGO@ZIF-L indicated a loosely stacked stratiform feature of the formed material.

The surface composition and element valence state of the composite were investigated using XPS. As shown in Figure 2A, the XPS survey spectrum indicated that the ERGO@ZIF-L composite contained C, N, O and Zn elements. The Zn 2p spectra contained two main binding energy peaks (Figure 2B), attributed to Zn 2p_3/2_ (1020.9 eV) and Zn 2p_1/2_ (1044.5 eV), which are characteristic of Zn^2+^ in ZIF-L [32]. Furthermore, the C 1s spectrum (Figure 2C) was deconstructed into four peaks at around 287.5, 286.1, 285.3 and 284.6 eV, which can be distributed to C=N, C-O, C-N and C-C, respectively [36]. The N 1s peaks (Figure 2D) were fitted to three peaks at 398.1 eV, 398.9 eV and 400.1 eV, and were related to C-N, C=N and N-Zn, respectively. Of these, the characteristic peaks at 398.1 eV and 398.9 eV were attributed to the characteristics of 2-MI [37], whereas the other peak was attributed to ZIF-L [32].

### 2.2. Electrochemical Characterization

The electroactive specific surface area of bare GCE (a) and ERGO@ZIF-L/GCE (b) were investigated using chronocoulometry (CC) (Figure 3A). The effective surface area (*A*) was calculated based on the Anson equation [38]:(1)$$Q\left(t\right)=2\mathrm{nF}A{\mathrm{cD}}^{1/2\;}t^{1/2}/\pi^{1/2}+Q_{\mathrm{dI}}+Q_{\mathrm{ads}}$$

In the Anson equation, Q_dl_ is a double-layer charge, Q_ads_ represents a Faradaic charge, and D is the diffusion coefficient. The electron transfer number represents n, substrate concentration represents c, and Faraday constant is represented by F. The curves of t^1/2^ with the Q (Figure 3B), A of bare GCE (a′) and ERGO@ZIF-L/GCE (b′) were calculated to be 0.078 cm^2^ and 0.486 cm^2^, respectively. This indicates that the ERGO@ZIF-L composite effectively increased the effective surface area of the electrode, thereby improving the detection sensitivity.

Electrochemical Impedance Spectroscopy (EIS) was conducted to evaluate the interfacial properties of the modified electrodes. The diameter of the semi-circle represented the electrode resistance arising from the charge-transfer resistance (R_ct_). The Nyquist plot shows the EIS of bare GCE (a), ZIF-L/GCE (b), ERGO/GCE(c), and ERGO@ZIF-L/GCE (d). Figure 4A shows that the R_ct_ value of ZIF-L/GCE (872.4 Ω) was significantly greater than bare GCE (316.2 Ω). This phenomenon was mainly due to the inherent poor conductivity of ZIF-L. However, the ERGO/GCE (131.6 Ω) possessed high conductivity compared to the bare GCE (316.2 Ω), attesting to the superior electrical conductivity and high electron transfer rate of ERGO. Notably, the R_ct_ value of the ERGO and ZIF-L composite was further decreased to 36.7 Ω. The good conductivity of the ERGO@ZIF-L composite can be ascribed to the synergistic effect between ERGO and ZIF-L, whereby the ERGO accelerates the electron transfer rate and the leaf-like ZIF-L works to produce a short electronic transmission path.

### 2.3. Electrochemical Behavior of BM

The electrochemical behaviors of BM (10.0 μM) for different modified electrodes were explored via differential pulse voltammetry (DPV). Figure 4B shows that no DPV peak was observed at bare GCE (a), which was due to the slow electron-transfer kinetics. A clear peak BM current response was observed at ZIF-L/GCE (b) and ERGO/GCE (c), which was due to the high-efficiency enrichment of ZIF-L for BM and the outstanding electrocatalytic ability of ERGO. Furthermore, the BM response current was dramatically enhanced at ERGO@ZIF-L/GCE (d). This excellent performance can be attributed to the synergy between ZIF-L and ERGO, which has the unique characteristics of high electrocatalytic properties, plentiful active sites, a fast electron-transfer ability, and a strong BM-enrichment capacity.

### 2.4. Optimization of Analytical Parameters

To obtain an optimal detection signal for BM at ERGO@ZIF-L/GCE, an optimization of the experimental conditions (material volume, accumulation time, and pH value) was necessary.

#### 2.4.1. Impact of the GO@ZIF-L Volume

Figure 5A displays the impact of the GO@ZIF-L volume on the DPV response toward BM (10.0 μM). The response peak current continued to rise from 1.0 μL, achieving a maximum of 5.0 μL. However, as the material volume continues to increase, the response peak current plummets. This is because the thickness of the ERGO@ZIF-L on the electrode causes considerable resistance to electron transfer [39]. Therefore, 5.0 μL was selected as the optimal material volume for subsequent detection.

#### 2.4.2. Impact of Accumulation Time

The influence of accumulation time on the DPV response BM (10.0 μM) was investigated from 30 to 300 s (Figure 5B). As shown, the current response peak increased gradually as the accumulation time increased from 30 to 120 s. When the accumulation time was longer than 120 s, the response tended to be constant. This could be due to the BM adsorption on the electrode surface achieving saturation. Consequently, the best accumulation time was estimated to be 120 s.

#### 2.4.3. Impact of Buffer Solution pH

The pH impact on the electrochemical response of BM (10.0 μM) at ERGO@ZIF-L/GCE was investigated using a pH range from 5.0 to 9.0 (Figure 6A). The response peak current increased as the pH changed from 5.0 to 7.0, and the highest peak current was observed at pH 7.0. A sharp decrease in peak current was then observed at higher pH values from 7.0 to 9.0, which was due to the instability of BM in the alkaline media [40]. Therefore, a buffer solution with a pH of 7.0 was chosen for further electrochemical detection.

Furthermore, it was found that the oxidation peak potentials moved in a negative direction as the pH increased (Figure 6B), which suggests that the proton participates in the electrochemical reaction of BM. Moreover, a good linear relationship was obtained between oxidation peak potentials (E_pa_) and the solution pH with the linear regression equation of E_pa_ = 1.117–0.055 pH (R^2^ = 0.993). The slope of −55.0 mV pH^−1^ was near the Nernstian theoretical value (−59.0 mV pH^−1^), indicating that the same number of electrons and protons participated in the BM electrode reaction [8].

### 2.5. Electrocatalytic Mechanism

At present, two different mechanisms are utilized to assess the electrochemical processes of Benomyl [8,9]. One is a two-proton and two-electron process. The other is a four-proton and four-electron process. To understand the oxidation mechanism and determine the number of electrons involved in the redox process, the cyclic voltammetry (CV) of BM (10.0 μM) at ERGO@ZIF-L/GCE with different scan rates was investigated (Figure 7A). As shown in the figure, there is a positive relationship between redox peak currents and scan rate (10–300 mV/s), and the redox peak potential gradually shifted towards positive potential. As exhibited in Figure 7B, a linear relationship was observed between the scan rate (*v*) and current redox peak (I_pa_ and I_pc_). The regression equation can be expressed as I_pc_ = 0.474 − 0.114 *v* (R^2^ = 0.998) and I_pa_ = −0.835 + 0.226*v* (R^2^ = 0.999). These relationships indicate that the redox reaction of BM at ERGO@ZIF-L/GCE is an adsorption control process [41].

In addition, the redox peak potential (E_pa_ and E_pc_) had a linear correlation with the logarithm scan rates and the regression equations of E_pa_ = 0.725 + 0.0368 ln*v* (R^2^ = 0.999) and E_pc_ = 0.837 − 0.0289 ln*v* (R^2^ = 0.990) (Figure 7C). This was based on Laviron’s equation [16]:(2)$$E_{\mathrm{pa}}={\;E}^{0}+\frac{\mathrm{RT}}{\alpha \mathrm{nF}}\ln\left({\mathrm{RTk}}_{s}\right)+\frac{\mathrm{RT}}{\alpha \mathrm{nF}}\ln v$$

Consequently, the electron transfer number (n) and electron transfer coefficient (α) can be calculated as 2 and 0.427, respectively. Since the electron value was equal to that of the proton (Section 2.5), the electro-oxidation of BM on ERGO@ZIF-L/GCE was calculated as a two-proton and two-electron process (Figure 1).

### 2.6. Determination of BM

In the optimum condition, the electrochemical performance of BM on ERGO@ZIF-L/GCE was evaluated using DPV (Figure 8A). As expected, the oxidation peak current increased with the increment of BM concentration (C_(BM)_) The oxidation peak current (Ip_(BM)_) presented good linearity in the concentration of the BM from 0.009–1.0 μM and 1.0–10.0 μM (Figure 8B). The linear relationship can be represented as I_p(BM)_ = 13.86C_(BM)_ − 0.0023 (R^2^ = 0.995) and I_p(Ben)_ = 5.40C_(Ben)_ + 5.825 (R^2^ = 0.997), respectively. The limit of detection (LOD) was calculated as 3.0 nM (S/N = 3), which exhibited a lower detection limit than most previously reported BM sensors (shown in Table 1).

### 2.7. Repeatability, Reproducibility, Stability, and Anti-Interference

The reproducibility of ERGO@ZIF-L/GCE was evaluated through 15 successive assays of 10.0 μm BM using one electrode (Figure 9A). The relative standard deviation (RSD) of the DPV response was 4.1%, suggesting th acceptable repeatability of the sensor. The reproducibility of the proposed sensor was evaluated by measuring the peak BM current (10.0 μM) using five electrodes that were prepared under similar conditions (Figure 9B). An RSD of 2.9% reveals the excellent reproducibility of the modified electrode.

Furthermore, the long-term stability of the sensor was evaluated by monitoring the BM (10.0 μM) every two days using a single electrode. From Figure 9C, it can be seen that more than 95.3% of its initial response was retained after 21 days’ storage, indicating satisfactory long-term stability.

Anti-interference ability is a vitally important factor for sensors; therefore, anti-interference studies were carried out during the experiment. The DPV responses of BM (10.0 μM) were measured without interfering substances and common coexisting substances. From Figure 9D, it can be seen that 80-fold common ions (Zn^2+^, K^+^, Mg^2+^, Na^+^, SO_4_^2−^, Cl^−^, and NO_3_^−^) and 30-fold pesticides, including carbendazim (CBZ), thiabendazole (TBZ), and dichlorophenol (DCP), did not influence the responsive BM current signal.

### 2.8. Real Sample Analysis

To validate the feasibility of the preparation sensor in practical applications, the presence of BM in tomatoes was detected. The sample pretreatments are explained in Section 3.4**.** The BM recovery experiments were performed using standard addition methods in the tomato sample, and the level of recovery was calculated from the calibration curve in Figure 8B. Typically, BM with different concentrations (0, 0.5, 3.0, 5.0 and 10.0 μM) were spiked separately into the tomato samples and tested using ERGO@ZIF-L/GCE. The recovery of Ben was in the range of 95.33–104.8%, with RSDs of less than 4.34% (Table 2). The analytical results indicate the utility of the sensor for detecting BM in a real-world sample.

## 3. Materials and Methods

### 3.1. Materials and Reagents

2-Methylimidazole (2-MI) and zinc nitrate hexahydrate (Zn(NO_3_)_2_·6H_2_O) were acquired from Aladdin Chemistry Ltd. (Shanghai, China). Graphene Oxide (GO) was provided by Nanjing XFNANO Materials Tech Co., Ltd. (Nanjing, China). *N*, *N*-dimethylformamide (DMF) was purchased from Aladdin (Shanghai, China). Benomyl (BM) was purchased from Macklin Reagent Co., Ltd. (Shanghai, China). Phosphate buffer solution (PBS) with various pH values was prepared from Na_2_HPO_4_ and NaH_2_PO_4_. All chemicals were used without any purification.

### 3.2. Apparatuses

The surface morphology analyses were obtained via scanning electron microscopy (SEM, Nova Nano450, Thermo Fisher, Waltham, MA, USA). The chemical composition and phase structure of materials were examined using X-ray diffraction (XRD, D8-Advance, Berlin, Germany). X-ray photoelectron spectroscopy (XPS, Escalab 250Xi, Thermo Fisher, Waltham, MA, USA) was used to investigate the elemental composition of composite materials. Raman spectroscopy with a laser wavelength of 532 nm was used (Horiba HR-800, Paris, France). Electrochemical experiments were implemented using an electrochemical workstation (CHI760E, Chenhua, Shanghai, China) with a three-electrode carbon glass electrode system (working electrode, GCE, ϕ = 3 mm), a platinum sheet (counter electrode), and a carbon glass electrode (working electrode, GCE, ϕ = 3 mm).

### 3.3. Preparation of GO@ZIF-L

ZIF-L was synthesized using Zn(NO_3_)_2_·6H_2_O as the metal source and 2-MI as the ligand [40]. In brief, 2.5 mmol of Zn(NO_3_)_2_·6H_2_O was dispersed into 15 mL of ultrapure water and dispersed with magnetic stirring. In another vessel, 25.0 mmol of 2-MI was dissolved in 90 mL ultrapure water and sonicated for 15 min. Subsequently, the two solutions were mixed and stirred for 8 h continuously, creating a milky white suspension. Next, ZIF-L nanosheets were collected by centrifugation and washed with ethanol several times; the resulting product was dried overnight at 65 °C.

The GO@ZIF-L composite was prepared using an ultrasound-assisted method. The resulting ZIF-L nanosheets (8 mg) and 4 mg GO were dispersed into DMF (2 mL) via sonication for 10 min to form a uniform solution. Subsequently, the ZIF-L suspension (4 mg/mL) and GO (2 mg/mL) dispersion were mixed for 2 h under vigorous stirring to obtain GO@ZIF-L. The resulting products were gathered by centrifugation and thoroughly washed with ultrapure water; the resulting product was then dried at 50 °C for 6 h.

### 3.4. Preparation of Modified Electrodes

Prior to the electrode modification, GCE was polished with Al_2_O_3_ powder (0.3 µm), followed by ultrasonication in deionized ethanol and deionized water. Next, 5.0 μL GO@ZIF-L dispersion (1.0 mg/mL) was coated on bare GCE and dried under an infrared lamp to obtain GO@ZIF-L/GCE. Subsequently, the GO@ZIF-L was reduced to ERGO@ZIF-L in PBS (pH 7.0) at a potential (−1.2 V) for 280 s using an i-t method. The modified electrode was expressed as ERGO@ZIF-L/GCE. Moreover, the ERGO/GCE was fabricated using the same i-t method, where 5.0 μL GO suspensions (1.0 mg/mL) were cast on bare GCE and dried under an infrared lamp. Subsequently, GO/GCE was immersed in PBS (pH 7.0) and reduced at a potential (−1.2 V) for 280 s using an i-t method to obtain ERGO/GCE.

ZIF-L/GCE was obtained by casting 5.0 μL ZIF-L suspensions (1.0 mg/mL) on bare GCE and drying these under an infrared lamp. The fabrication and detection process of the sensor is illustrated in Figure 2.

### 3.5. Preparation of Practical Samples

To conduct a real-world analysis of vegetable samples, tomatoes were purchased from the local food market. A piece of fresh tomato sample (around 10 g) was crushed to homogenate using a juicer and then separated by a 0.22 mm membrane filter. Individual sample solutions were prepared by diluting the tomato samples 50-fold with PBS (pH 7.0).

## 4. Conclusions

In this work, the ERGO@ZIF-L nanohybrid was prepared with facile ultrasonic blending and an electrochemical reduction method. The nanohybrid was further used to design a sensing platform for the detection of BM. The synergistic effect between ERGO and ZIF-L providees ERGO@ZIF-L with its high electrocatalytic properties, plentiful active sites, fast electron-transfer ability, and a strong capacity for enriching BM. As expected, the proposed sensor exhibited an outstanding electrocatalysis performance for BM with a low-limit detection of 3.0 nM. More importantly, the sensor also possessed high selectivity, excellent stability, good reproducibility, and acceptable recovery in a real-world sample, indicating that ERGO@ZIF-L is a promising candidate for electrode material.

## Data Availability

The data presented in this study are available in the article.

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
