# Peer review of "2D Leaf-Like Structured ZIF-L Embedded Electrochemically Reduced Graphene Oxide Composite as an Electrochemical Sensing Platform for Sensitively Detecting Benomyl"

_molecules, 2022, doi:10.3390/molecules27206857_

Round 1
Reviewer 1 Report
In this paper, the authors have reported a new platform as a sensor for the detection of benomyl. The platform is based on a composite of ZIF-L MOF with reduced graphene oxide. The manuscript is well presented and the results are interesting. However, there are some points that should be improved before publication.
1. There are some typos and grammatical mistakes in the manuscript. The authors should revise the English of the manuscript and correct the typos and grammatical mistakes.
2. What is the advantage of the electrochemical reduction of graphene oxide? The authors should explain the advantage of electrochemical reduction of graphene oxide and explain why they have emphasized this method for the reduction of graphene oxide.
3. At the beginning of the second paragraph of the introduction section, the authors noted the significance of the surface reaction kinetics and supported their claim by citing a paper. The cited manuscript is not specifically discussed the kinetics of the surface reactions and only presents an example of modification of the electrode for sensing application. The authors should support their claim with more appropriate and relevant citations.
4. In the introduction, the authors should present more details of the application of MOFs in general, and ZIF-L in specific to show how their properties, including surface area, conductivity, etc affect their behavior in electrochemical sensing applications.
5. Raman spectra are required for the characterization of ERGO@ZIF-L. The Raman spectra of ERGO@ZIF-L and ERGO should be added and compared.
6. How the authors have characterized ERGO@ZIF-L, while it is synthesized on the surface of GCE? It should be clearly explained.
7. The fabrication of ZIF-L/GCE and ERGO/GCE that are used in the manuscript should be presented or cited with appropriate citations.
8. In section 2.4.1, how is the use of different volumes of ERGO@ZIF-L? What is the volume in microliters, while the electrode is solid?
9. Why DPV responses are lower in acidic and basic conditions?
10. In section 2.5, BM (10.0 um), what is μm?
11. In section 2.5, the authors have presented the possible electrocatalytic mechanism. However, the suggested mechanism is not well presented. The authors should discuss more the mechanism of the reaction and compare it with other possible mechanisms. Finally, the authors should support their claims of which type of mechanism is better in agreement with the results. This part needs very major revision.
12. In Scheme 1, how do the authors know the product of the reaction? Have they characterized the products? They should purify and characterize the product, or cite suitable citations. The authors can see https://doi.org/10.1016/j.elecom.2019.05.023, which has purified and characterized the product of the reaction.
13. In section 3.3, in Zn(NO3)2·6H2O the numbers 3,2, and 2 should be subscripts.
14. In section 3.4, what is the concentration of GO@ZIF-300 L dispersion?
Author Response
Please see the attachment。

Reviewer 2 Report
In the present manuscript, the authors have studied 2D Leaf-like structured ZIF-L embedded electrochemically re- 2 duced graphene oxide composite as an electrochemical sensing 3 platform for sensitively detecting Benomyl 4. The findings are ok but I have few concerns about the work discussed. My observations about the manuscript are:
1. The abstract is two weak and does not reflect the finding of this work, abstract should be revised.
2. The introduction lacks strong motivation behind the work as several other similar reports are already available hence comparatives studies should be added.
3. English editing is a must, hence the manuscript should be thoroughly revised for grammatical/typo errors
4. Equation 2 should be type in equation mode.
5. During preparation of Practical sample, sample solution was prepared by diluting 50 -fold of the tomato sample. Account for the reasons for selsction of this fraction.
6. On what basis did the author choose the portion of Zn(NO3)2·6H2O for analysis?
7. Conclusion should be made elaborate.
Round 2
Reviewer 1 Report
Dear Editor,
I see the authors have revised the manuscript and believe it could be accepted for publication in its current form.